# Effectiveness of New Reactivation Approaches in Integrated Long-Term Care—Contribution to the Long-Term Care Act

**DOI:** 10.3390/healthcare13101187

**Published:** 2025-05-19

**Authors:** Suzanna Mežnarec-Novosel, Marija Bogataj, David Bogataj, Eneja Drobež

**Affiliations:** 1Department of Social Gerontology, Alma Mater Europaea University, Slovenska 17, 2000 Maribor, Slovenia; marija.bogataj@guest.arnes.si (M.B.); david.bogataj@almamater.si (D.B.); eneja.drobez@us-rs.si (E.D.); 2Institute INRISK, Kidričeva 1, 8210 Trebnje, Slovenia; 3Constitutional Court, Beethovnova 10, 1000 Ljubljana, Slovenia

**Keywords:** gerontology, older adults, legislation, home care, integrated care, reactivation, physiotherapy, occupational therapy, Z + 4 test, effectiveness, EU projects

## Abstract

**Objectives:** This study evaluates the effectiveness of an innovative, integrated healthcare approach based on the “Long-term Care (LTC) in the Community” pilot project, aimed at testing solutions for the new LTC Act and associated regulations in Slovenia. It is based on a small sample, which can be financed from European project funds. This methodology is crucial for the subordinate legislation derived from the LTC Act. **Methods**: We selected beneficiaries who decided to receive integrated services in their home environment in rural areas. Among 132 beneficiaries who received various care services at home, 75 beneficiaries received integrated services to maintain independence, while a control group of 57 persons did not receive these despite eligibility. The hypothesis that the new reactivation and other services to strengthen and maintain independence facilitate a transition to a lower category of LTC within the system of different states of abilities and that new approaches with integrated home care are much more successful were tested using the Z + 4 test. **Results**: Seven out of eight users’ abilities were improved by an inventive approach to LTC at home, including reactivation activities, with *p* < 0.025. The probability that the new methods will outperform the old ones for an arbitrarily selected senior eligible for LTC exceeded 0.89. Thus, we used quantitative methods to confirm the reasonableness of the decisions included in the LTC Act and will help to estimate savings in the health fund. **Conclusions**: These positive results promote the implementation of integrated care strategies, norms, and standards, which will be further articulated in subordinate legislation.

## 1. Introduction

### 1.1. Longevity, Which Requires Greater Care for the Elderly Population

Longevity is an achievement of developed countries. The resulting changes in the population’s demographics require a careful plan on how to provide care for senior citizens in local communities. Given their limited financial resources, decision-makers are more inclined to select services with quantifiable and observable results. A study by the European Commission forecasted a rapid growth in spending on long-term care (LTC) within the European convergence scenario, not only due to demographic changes but also due to the public financing of formal services not covered by previous public LTC programmes [1]. This fact highlights the necessity of prioritising the growth of community-based care services over the expansion of institutional care capacity, as stated in the European Commission’s The Green Paper on Ageing [2].

Nearly three-quarters (73.2%) of those 85 years of age or older in Europe (EU-27) reported having a chronic illness or health issue, according to Eurostat [3]. This proportion declined with age. About two-thirds (65.7%) of those aged 75–84 had a long-term disease or health issue, but the comparable percentage for those aged 65–74 was lower (54.8%). Among these, bed immobilisation can hasten functional decline or impair functional capability, increasing health vulnerability, including the risk of falls and the chance of developing new health issues. Older adults are frequently bedridden because of disease or injury. Frailty, a syndrome of functional, cognitive, and social impairments that often coexists with multi-morbidity and a lower quality of life, is a significant problem associated with ageing [4]. One out of every three people 65 and older falls yearly, most often at home. The critical factors associated with falls are severity and mobility and balance problems, bladder dysfunction, fear of falling, fatigue, and cognitive dysfunction [5]. According to the World Health Organization [6], falls constitute the second primary cause of unintentional injury fatalities globally. Annually, around 684,000 individuals succumb to falls worldwide, with more than 80% occurring in low- and middle-income nations. Individuals above the age of 60 experience the highest incidence of fatal falls. Each year, 37.3 million falls necessitate medical attention. Consequently, the WHO recommends that preventive methods enhance education and training while facilitating the establishment of safer settings. The WHO also recommends prioritising fall-related research and implementing effective regulations to mitigate risk. Consequently, seeking methods to reduce fall risks is prudent, as it can improve the environment and services, thereby reducing health insurance expenses and enhancing the quality of life for older adults. Therefore, these aspects should be considered for prevention and activities after an incident when planning the LTC.

### 1.2. The European Union’s Response to the Phenomenon of Rapid Population Aging

Through various programmes like Ambient Assisted Living (AAL), Horizon, and Interreg, as well as through European funds like the European Social Fund and the Regional Development Fund, the European Commission has supported numerous projects in this quarter of a century to lower the growth of public expenditure and improve the quality of life for older adults over a longer time horizon. These projects aim to develop and test social innovations to reduce costs or expenses and to include the findings in the LTC Act and its subordinate legislation. Funding social innovations that have positive social impacts is one of the primary goals of national and international cooperation initiatives [7]. However, integrating innovations often requires not only the implementation of individual projects but also systemic changes in the understanding of ageing, care, and the responsibilities of the public and private sectors. It is especially crucial for nations that are only working on standardised LTC systems to incorporate effective advanced care for older adults. Slovenia is also one of them. Social innovations prioritising teamwork, empowerment, and moral purpose are in a strong position to dispel antiquated beliefs and promote inclusive, long-term care solutions. Alternative model frameworks are essential to innovation development and viability because they lessen bureaucratic restrictions and encourage integrated care. Legislation and finance arrangements also significantly impact these implementations, as Rothgang [8] contends.

In Slovenia, older individuals unable to attend to their daily needs have historically benefited from various social systems, ranging from familial caregiving to institutional support in elderly homes and nursing care facilities. Due to rising life expectancy, a growing cohort of older adults, the geographical distances of children from their parents, and other factors, society needed to establish new LTC programmes and an LTC insurance system. In 2018, the Ministry of Health of the Republic of Slovenia issued a tender for pilot projects to facilitate the enactment of the LTC system. Three pilot environments (urban, semi-rural, and rural) were chosen based on population density to test integrated services in the case of home care. The Municipality of Krško successfully applied for the project and created a test environment in rural areas as part of the LTC in the Community MOST (Bridge) project, where the first author was the project manager. Fieldwork enabled the collection of a large amount of data, including numerical data, but the final reports were more focused on qualitative analyses and less on how much the new approach with integrated home care could contribute to increasing the functional capabilities of the residents and, thus, to reducing the costs of the LTC or health insurance fund.

Given the limitations of public budgets, assessing the social system’s financial sustainability is essential, which cannot be achieved without in-depth and sufficiently accurate quantitative analysis. Consequently, the article aims to illustrate the methodology for assessing the data required to evaluate the effectiveness of integrated LTC services when contemplating an innovative home care approach incorporating new service types, even with a reduced sample size, as exemplified by the MOST experiment funded by European projects in Slovenia. Specifically, this approach should also be included in the subordinate legislation derived from the LTC Act.

We have dedicated our article to addressing this question. From the second section onward, we explore insights from the foreign literature on the topic. In the first subsection of the third section, we introduce the development of long-term care, with a particular focus on integrated care within the country and including a description of the collected data, which had not yet been quantitatively processed within the MOST project. This is followed by a more in-depth quantitative analysis, presenting new results along with a discussion and proposed guidelines for future research.

## 2. Literature Review

### 2.1. On the Effectiveness of Innovative Approaches to Long-Term Care

Research on the impact of individual interventions on the effectiveness and sustainability of LTC systems often examines home-based and institutional care separately. When introducing new policies in this area, experiments are needed to evaluate individual interventions, compare these two types of care, and improve knowledge about LTC integration. Reviews of experiments conducted within LTC projects mainly provide descriptions of qualitative effects on the users’ quality of life included in the trials [9,10,11]. Fewer experiments in LTC include quantitative impact assessments, such as the article by López-Torres Hidalgo [12] or Carron et al. [13], and cost-effectiveness of an integrated LTC service or programme, such as presented by de Batlle et al. [14]. For example, within the Horizon 2020 CONNECARE project in Catalonia, Spain, the latter showed that an integrated care programme experiment generated savings under different cost scenarios [15]. They calculated the cost-effectiveness ratio (ICER).

Experiments in London aimed at integrating primary, acute, community, psycho-logical, and social care revealed that the evaluation process could be hindered by the absence of a definitive evaluation strategy, thereby impeding the acquisition of evidence necessary to substantiate claims regarding the advantages of integrated care [16]. The research under the UHCE project (Urban Health Centres Europe) concentrated on effect and process evaluations employing a multi-level modelling methodology [17]. These and several other instances of qualitative impact assessments of experiments can enhance the findings of quantitative studies; nonetheless, they cannot fulfil the requirement to validate the sustainability of systemic solutions for which management policies would be established.

Those involved in developing social systems need reliable and precise evidence to substantiate the effectiveness of particular solutions and ascertain which choice produces the most favourable outcomes while functioning within limited financial resources [18]. This issue is urgent in all branches, as engineers provide technical solutions, while the health sector and financiers utilise differing criteria to evaluate the effectiveness of these solutions. Engineers evaluate technological functionality, the healthcare industry relies on clinical trials for assessments, and financiers concentrate on the prospective reductions in public spending produced by new services utilising novel methods, approaches, and technologies established in projects.

This article aims to illustrate the methodology for assessing the data required to evaluate the effectiveness of the new integrated LTC services even with a reduced sample size.

### 2.2. Selected Integrated LTC Services Described in the Literature Review

To reduce health insurance expenses, hospitals are reducing the number of hospital days after surgeries and other treatments. Therefore, acute care patients are increasingly being discharged from hospitals in the home care provided by community nurses and social workers [19,20]. This may be attributed to modern surgical techniques and the move towards a more economically sustainable health system [21]. As a result, more and more older adults, including those with complex health conditions, are being treated in community care settings [22]. In addition, it is essential to consider the potential adverse effects of discharge, including feelings of abandonment, anxiety, family conflict, and distrust [23], which commonly undermine the effectiveness of care delivery. Health and social care should equip patients to live in their home environment. The primary goal of postoperative care, together with rehabilitation and reactivation services, is to facilitate the patient’s maintenance or improvement of physical function, quality of life, and rapid achievement of independence, all of which are provided not only for the very old but also for younger people discharged from hospitals [24]. Consequently, a holistic approach to home and community-based care is necessary [25]. When examining the importance of reactivation in LTC, it becomes apparent that German social security regulations can be a role model for us, primarily due to the strong endorsement of the principle of rehabilitation before and during care (Rehabilitation vor und bei Pflege), as articulated by Küpper-Nybelen et al. [26] and the current content of the Federal Ministry of Health [27]. This principle is essential for mitigating the risk of falls and frailty in older adults, allowing them to remain in their home environment longer.

To obtain information on the effectiveness of new therapies in long-term care in other care systems worldwide, we searched the authors’ findings in the Web of Science (WoS) Core Collection. Articles were selected on 12–13 February 2025, covering publications from 1990 to December 2024. They examine the effects of the newly selected services. The flow chart in Figure 1 depicts the trends in article publication within journals indexed in the WoS Core Collection. It shows, outlined in keywords by merging the search terms “long-term care” (LTC) and “home care” (HC), each of the following keywords (the new selected therapies) separately: physiotherapy, occupational therapy, kinesiotherapy, rehabilitation, reactivation, reablement, and restorative.

The data show that in the field of research in LTC, the term rehabilitation is most often used. Kinesiology has the most potential for new research. Figure A1 and Figure A2 in Appendix A show the dynamics of article postings after individual therapies in LTC and HC Long-Term Care. The literature review also confirms that physiotherapy is rarely considered in home care research. We expected that the term reactivation would be used more frequently in articles, as we used it for a set of therapies that contribute to the reactivation of people in LTC. When comparing the data from the graphs and the content of the articles, it was observed that the terms reablement, restorative care, reactivation, and rehabilitation were often used interchangeably but had specific differences, especially in the context of long-term care and health services for the elderly. Sims-Gould et al. [28] describe the 4R terms as intensive, short-term programmes designed to help older adults regain or maintain the ability to perform some aspects of their care independently. Gough et al. [29] note that these approaches lack precise definitions, making their comparison and implementation difficult.

As the diagram shows, we shortlisted 67 contributions by excluding duplicates and articles based on content. We concentrated on home care services’ advantages and economic efficiency, primarily accessible in rural regions. Cook et al. [30]; Anthun, Lillefjell, and Anthun [31]; and Song, Lin, and Hung [32] demonstrate that research frequently amalgamates occupational therapy and physiotherapy, highlighting the substantial design, adaptation, and individualised characteristics of the 4R interventions (rehabilitation, restorative, reactivation, and reablement). However, specific quantitative outcomes are notably absent.

Among 67 publications in journals indexed in the WoS Core Collection and counted in Figure 1, only six address the particularities of rural areas. They do not examine the integrated methodologies. The reviewed articles reveal the disparities in the availability of adequate home care services between urban and rural care networks [33]. Consequently, rural residents exhibit an increased likelihood of opting for institutional care [34] or experiencing a heightened risk of falls [35], further impacting the shift to institutional care. The article by Cook et al. [30] asserts that prior in-home physiotherapy significantly mitigates the risk of needing to go to institutionalisation following short-term hospitalisation. Consequently, in our case study, physiotherapy as part of the services to maintain independence was particularly highlighted when examining advances in home care.

Integrated home care, encompassing physiotherapy, nursing, occupational therapy, social work, kinesiology, and related disciplines, necessitates an increased number of experts and, thus, incurs elevated travel expenses for caregivers. The treatments are expensive but may reduce insurance costs over time [36]. Incorporating diverse professional profiles in the case of home care is crucial, particularly when managing an individual transitioning from a hospital to a home setting [37]. There is often a need for health support at home, advice for housing environment adaptations, and a need for help due to a feeling of abandonment, anxiety, decreased ability to socialise, etc., which dictates comprehensive social and health care. In those cases, we discuss reactivation approaches [28] to improve functional abilities and capabilities. An increase in functional abilities means that the assessment of these capabilities, described by points, is lower. The reactivation programmes are among the most essential and effective for enhancing functional abilities and capabilities [38].

An increasing number of studies demonstrate that integrated and interdisciplinary approaches retain and enhance functional capacities in older adult cohorts [38,39,40,41] and that physical medicine and rehabilitation in the home environment could be equivalent to rehabilitation in geriatric and other institutions [42]. Some authors exposed that physiotherapy can stimulate older adults, primarily through different reablement programmes in restorative care, which are typical for activities in the United Kingdom [31], Australia [43], and Germany [27], known as the “Rehabilitation before long-term care and during care” approach whose mission is to prevent or reduce the need for LTC.

Also, residents’ perceptions regarding recreational physiotherapy are highly positive. Surprisingly, physiotherapy can also be essential in managing malnutrition in older adults [44]. According to Linhares et al. [45], practitioners must consider residents’ perceptions of this practice when developing and implementing physiotherapy in long-term care; thus, Person-Centred Care proves effective for older adults, benefiting and satisfying relatives and other informal caregivers [46]. The authors suggest various exercises, like chair-based resistance band exercise, significantly promoting daily living activities, handgrip strength, limb muscle endurance, body flexibility, breathing exercises, and dynamic balance, when including physiotherapy, kinesiotherapy or occupational therapy in integrated care [47,48], but only suitable and timely rehabilitation interventions yield optimal health and social integration outcomes [49].

Decision-makers need to create clear eligibility criteria regarding the categorisation of care, assess efficacy accordingly to provide individuals with appropriate long-term care services, and introduce them into the legislation. The literature evaluation lacks explicit criteria for identifying and statistically assessing optimal processes, with impacts and usage recommendations based on quantitative and not only qualitative criteria. The subsequent sections present an example of evaluating innovative home care solutions implemented in Slovenia’s LTC pilot projects financed by the European Social Fund and the Republic of Slovenia.

## 3. Methods and Models Supporting the Long-Term Care Act

### 3.1. The Long-Term Care Act

Slovenia’s accession to the European Union in 2004 brought new frameworks for harmonizing social policy and spurred reforms aligned with broader European social protection standards. However, Long-Term Care, compared to pension reforms or healthcare restructuring, was not initially a legislative priority despite demographic projections indicating rapid population aging. The intention to regulate better LTC and finance it within the social insurance framework has been discussed in Slovenia for nearly 20 years. The first legislative proposal was prepared in 2006 and improved in the next 17 years by the Ministry of Health and the Ministry of Labour, Family, Social Affairs and Equal Opportunities. Finally, the Parliament adopted the proposal by the Ministry of Health in 2021 as the first LTC Act [50]. After several modifications, the new LTC Act was adopted in 2023 [51], with further amendments in 2024 [52]. The timeline for implementation of the rights includes the right to the integrated long-term care at home and e-care starting on 1st July 2025 and the right to long-term care in an institution or cash benefits (and the possibility of substitute care within the framework of the right to a family caregiver) on 1st December 2025.

As a novelty, the LTC Act [51] also includes services for strengthening and maintaining independence, consisting in particular (a) services aimed at strengthening and maintaining functional abilities and developing substitute skills to increase and maintain independence; (b) consulting services for adapting the living environment; (c) post-diagnostic support services for persons with physical or mental health problems and support for their relatives. The intensity and duration of these activities have been quantitatively determined so that an increase in functional capabilities means that the assessment of these capabilities, described by points, is lower. However, standards and norms for human resources performing these activities have not been defined yet.

The LTC Act [51] places the individual at the centre of the consideration and primarily aims to maintain its independence and adapt the care method to the individual care recipient. The Act includes all adults (18+) in the long-term care system. The core of this new vision of long-term care is to strengthen the community’s network of services, enabling all those who need long-term care to remain in their home environment for as long as possible. At the same time, the goal of the reform is to ensure that services for those who need institutional care will also be more financially accessible and achievable, even within the scope of the infrastructure available today.

Article 12 LTC Act [51] defines the LTC category of care as the range of rights to which the insured person is entitled based on the eligibility assessment as follows: Category 1—mild limitation of independence or ability to care for oneself;

Category 2—moderate limitation of independence or ability to care for oneself;

Category 3—severe limitation of independence or ability to care for oneself;

Category 4—more severe limitation of independence or ability to care for oneself;

Category 5—the most severe limitation of independence or ability.

To properly provide care in each category, it is necessary to plan new adapted forms of community services, train the required human resources, and ensure appropriate social infrastructure [53]. Financial expenditures for LTC underscore the necessity of statistically monitoring and forecasting the population requiring health and social care by category when strategizing enhancements in home care and other care organisation. Transitions among various forms of community care are associated with numerous demographic, health, and regional characteristics that influence older individuals’ abilities for self-care, subsequently affecting the financial sustainability policies related to population ageing and increasing public expenditures on long-term care.

Therefore, to introduce better methods into the LTC system, it was necessary to review the foreign literature and foreign experiences in more detail before the new legislation was implemented. Thus, based on this, pilot projects were carried out in three Slovenian municipalities: rural, semi-urban, and urban areas. We present a study that was carried out in a rural area. The results have been implemented in the new LTC Act, and further consideration will be given to optimally incorporating them into subordinate legislation.

### 3.2. Study Design

#### 3.2.1. MOST Project as an Innovative Approach to the LTC

In 2018, the Ministry of Health of the Republic of Slovenia issued a tender for pilot projects to examine the possibility of introducing new services in long-term home care [54]. Three pilot environments (urban, semi-rural, and rural) were chosen based on population density to test integrated services in the case of home care. The Municipality of Krško successfully applied for the project implementation under the acronym MOST, which commenced in January 2019 and concluded on 31 December 2020.

The project evaluated the comprehensive long-term care process, commencing with an application to be served by the user or a family member and concluding with the completion of service utilisation (Figure 2). A singular entrance point was developed for all candidates within the LTC network. A novel approach for evaluating eligibility was tried. Services to maintain independence, health services, and e-care have been added as options to the previously established services, as presented in Figure 2. This is discussed further in the subsequent sections.

The primary requirement was to facilitate a person-centred care approach. Diverse novel methodologies for collaboration and stakeholder integration were evaluated in pilot settings. The pivotal function of the LTC coordinator was validated in the pilot. The coordinator was responsible for facilitating effective communication and coordinating activities among local community stakeholders from the health sector, including doctors, nurses, and social workers in hospitals, personal physicians, and visiting community nurses, as well as social care providers from the Social Work Centre, employees at the Single Entry Point, the Care Team (CT), the Independence Maintenance Team (IMT), informal caregivers, and volunteers. The coordinator was tasked with formulating and updating users’ individualised plans, organising care, coordinating schedules, and prioritising users’ requirements. Following the coordinator’s strategy to address the needs, educational and training programmes for informal caregivers and volunteers were organised. The project’s complexity and scope overloaded the coordinator and influenced staff turnover in these workplaces. These challenges highlighted the need to develop appropriate human resources standards and norms. The coordinator’s work contributed to better coordination of services, increased accessibility to care, and improved users’ quality of life. In the pilot project’s final phase, the effectiveness of the provided services was assessed (coloured blue in Figure 2).

As Figure 2 shows, integrated healthcare services and services for maintaining independence, which were previously provided only within institutional care or in rehabilitation centres and spas like physiotherapy, kinesiotherapy, occupational therapy, and newly designed psychosocial support services, which were not part of previous home care practices, have been treated as novelty integrated into home care (see also Table 1). Table 1 compares the current LTC system in Slovenia, as received by the control group in our study, and the system envisaged by the new legislation, which we evaluate in this article.

The new group of services for maintaining independence in the MOST pilot project consisted of as many as 16 different services, from psychosocial support to maintaining or increasing independence, which reduces the need for help from others. The independence maintenance unit comprises physiotherapists, occupational therapists, social workers, and master kinesiologists.

The pilot project evaluators used questionnaires to measure the effects of the integrated services. The evaluation report [55] contains extensive results demonstrating the pilot activities’ contribution to individuals’ quality of life. Still, we can conclude that they were not quantitatively oriented enough.

These reports provide the basis for designing solutions for the systemic implementation of long-term management at national, regional, and local levels. In this paper, we report on the quantitative measurement of new approaches’ impact on service users’ functional abilities, which were not known and presented in the evaluation report.

Pilot projects introduced several innovations in the LTC, which significantly contributed to developing a systemic approach and improving the LTC Act. In addition to integrated services, methods, and procedures, these innovations also included e-care presented by Mežnarec et al. [56], which is not the subject of our study.

#### 3.2.2. NBA Assessment Tools and Differences Introduced in the Slovenian LTC Act

Before the adoption of the first LTC act [50], several laws regulated various aspects of long-term care:

-Act on the Social Protection of Mentally and Physically Disabled Persons [57];-Health Care and Health Insurance Act [58];-War Veterans Act [59];-War Disabled Persons Act [60];-Acts on Social Benefits and Social Welfare Services [61];-Financial Social Assistance Act [62];-Exercise of Rights to Public Funds Act [63];-Pension and Disability Insurance Act [64];-Parental Care and Family Benefits Act [65].

The respective services were divided into six social security subsystems that were not well connected.

Various scales have been employed to evaluate applicants’ needs. Slovenia had to decide on a suitable instrument for determining eligibility for LTC services to prepare for writing the LTC Act. After examining several instruments used both domestically and internationally (in Austria, Germany, the Netherlands, and Finland), the working committee selected the German tool Das neue Begutachtungsinstrument—NBA [66,67,68] and adapted it to the Slovenian legal and practical environments.

In response to changing demographics and rising care demands, Germany modified eligibility requirements and contribution rates in 1994 and 1995 [69]. The second LTC reform, which went into effect in Germany in January 2017, included the NBA tool [67]. With the reform, Germany altered its qualifying criteria for long-term care services, emphasising the social and cognitive components of the recipients’ lives. The qualifying procedure and personalised service planning are rigorously separated in the German system, both methodologically and procedurally. Indicators, including comprehensiveness, methodological appropriateness, and provenance, played a role in Slovenia’s selection and modification of the German tool [68]. Its approach continues to serve as a model for other nations thinking about contributory finance strategies and is a prime example of a country with a sophisticated LTC social insurance system [70].

The revised assessment guidelines for Slovenia have been implemented in the pilot project MOST. The NBA tool assesses eligibility for LTC provision through eight modules (Table 2) and classifies persons into five care categories (Table 3) as a novelty in Slovenia. This approach is in the LTC Act defined in Art. 12 and Art. 36. After a detailed analysis of the NBA tool and a comparison with the characteristics of care and needs in Slovenia, the working group for preparing the bases for the LTC pilot project concluded that the German tool was required to be adapted to the Slovenian system and needs. NBA-SLO was the name given to this modified tool. The primary reasons for the alteration were the differences in social care practices and the existing infrastructure. The plan was to ensure that all relevant services were included in assessing LTC needs; hence, the NBA tool was revised. The subsequent chapter delineates the services in the NBA-SLO methodology ([51], Art. 36).

Slovenia incorporated the modified NBA tool into a procedure that consists of the NBA-SLO assessment with the user’s life narrative and a personal plan. The Slovenian model also highlights the value of individual planning and a more comprehensive strategy that integrates social and health viewpoints. A more profound comprehension of the person and their needs is made possible by knowing the life story. It contributes to a more customised evaluation. The Slovenian approach sought to use the NBA-SLO as a basis for individualised planning, in contrast to the German system’s strict separation of the eligibility procedure and personalised service planning.

Given the characteristics of Slovenia, it was essential to revise Modules 7 (activities outside the home environment) and 8 (ability to perform household chores in the environment where the insured lives). The adjustment led to a restructuring of the weights in the NBA-SLO model to maintain the sum of weight equal to 100 (Total of points in bold in the table), with solutions executed as illustrated in Table 2. In Germany, the data from Modules 7–8 are employed in developing personal care plans following the ninth [72] and the fifth [73] volume of the Social Code.

Following the rules on the evaluation scale for assessing entitlement to LTC [50], which was adopted in the framework of the LTC Act in Slovenia [51], the sum of the weighted points, as given in Table 3, for the presented modules is taken into account when determining the LTC category, whereby Module 2 and Module 3 lead to 15% of the total number of weighted points, whereby the module in which the insured person scored a higher number of weighted points is taken into account in the total number of points; Module 6 and Module 7 result in up to ten percent of the total weighted points, whereby the module in which the insured person scored a higher number of weighted points is taken into account in the total points.

Following the eligibility evaluation, in Germany, a personal plan is created for the beneficiary that outlines how the funds will be utilised to reach his objectives. The expenditures are based on the eligibility evaluation (care category) rather than the plan itself. In Slovenia, the personal plan is developed in collaboration with the user following the eligibility evaluation. This plan outlines the beneficiary’s life circumstances, objectives, and the particular services and resources required, as the personal plan plays a crucial role in the eligibility assessment process.

Table 2 outlines the evaluation method for functional abilities through modules featuring five levels of functional capacity and translate the points from each module into weighted point values within the NBA-SLO assessment tool. An increase in functional abilities means that the assessment of these capabilities, described by points, is lower, or that the patient has moved to a lower care category. Table 3 shows the weighting limits in the NBA-SLO tool, which are the basis for classifying a person into an LTC eligibility category. Based on the LTC Act [51], five care categories are used in Slovenia. The functional capacity threshold for treating a person within the framework of the LTC Act is 12.5 weighted points.

In our study examining the effectiveness of reactivation interventions in home care, we assess effectiveness based on individual abilities across modules (M1-M8). Each module is assigned a maximum corresponding point value: P(M1) = 10, P(M2) = 15, P(M3) = 15, P(M4) = 40, P(M5) = 15, P(M6) = 10, P(M7) = 10, and P(M8) = 10.

### 3.3. Determining the Sample for This Paper

The case study utilised data regarding long-term care services provided to beneficiaries in the home setting who were permanent residents of the municipality of Krško. From the initial evaluation of an individual’s eligibility for LTC services to the last assessment, or until the project’s conclusion, the longitudinal study tracked the states of people’s abilities assessed with the points listed in the last column of Table 3. It is, therefore, classified into one of the listed classes. Assessments were performed for two years contingent upon alterations in the users’ capabilities that could necessitate a modification in the care classification following the stipulations of the public tender.

During this period, 538 applicants were chosen. Applicants from institutional elderly homes, care work centres, and those who did not undergo an initial assessment (due to death, withdrawal from the personal plan, or transfer to institutional care before the first assessment) were excluded from the study. Consequently, the sample comprised only those assessed applicants deemed eligible for long-term care services in their residences (home care, n = 352). For persons in institutional care, the goal of the MOST project was not to include them in the integrated services but only to test the NBA-SLO assessment tool. At least two evaluations were required during the project implementation period. Beneficiaries (n = 220) who obtained assistance externally or who, for various reasons, underwent only a single assessment were excluded. Within the pilot project framework that supported the transition to the implementation of the LTC Act, 132 users were observed who received integrated LTC services from 16th January 2019 (the initial service supplied) to 23rd December 2020 (the last service provided in a home environment). Among these, 77 were female and 55 were male. The mean age of the participants was 78.3 years. The youngest beneficiary was 24 years old and the oldest 98 years old because the public tender specified the age of majority (persons 18+) as an entry requirement.

### 3.4. Statistical Analysis

#### 3.4.1. Participant Selection

In this statistical analysis:

-A total of 132 home care beneficiaries who consented to integrated care were chosen in the random sample.-Furthermore, 57 eligible individuals, with a mean age of 79.8 years, who declined this novel therapy, were selected as a control group.

Eight components were employed to assess the efficacy of integrated long-term care, as detailed in Table 2. Each module was assigned a specific weight, expressed as a percentage. The abilities and capabilities of each beneficiary were assessed within each module. The recipient was assigned to the care category that aligns with his functional abilities (FA) as determined by the overall assessment results. An increase in functional abilities here means that the assessment of these capabilities, described by points, is lower or that the patient has moved to a lower care category.

#### 3.4.2. Intervention Details and Assessment Tools

As described in Table 2, functional abilities were evaluated according to eight modules. In the overall assessment that resulted in the category of individual care, all services (16) presented in Table 4 were included.

The result was determined by subtracting the first assessment’s balance from the final evaluation’s ability level among interventions. The percentage of individuals with a positive difference in assessment $p_{i}=X_{i}/N_{i}$, at these two observations, where i=1 means those under integrated care and i=2 is the notation for the control group ($N_{2}$ = 57 and $N_{1}$ =132), was evaluated. Since the sample size is often small in pilot studies, we suggest using the Agresti–Caffo test for percentage testing [74]. Therefore, the Agresti–Caffo evaluation Z + 4 was used to assess the critical limits for the percentages of customers with enhanced abilities based on particular modules at *p*-value < 0.025. The final ability state was calculated by subtracting the balance from the last ability level assessment and the ability level of the first assessment. Using the Z + 4 test, we calculate the upper and lower bounds of the proportion of people whose evaluation of the individual module rises (being reassigned to a lower care category presented in Table 3). The critical limits for the percentages of clients with improved abilities according to individual modules were estimated with the Agresti–Caffo evaluation at *p*-value < 0.025.

When estimating the upper and lower limits of the percentage of positive ratings, we considered that the posterior distribution of the percentage of positive ratings is distributed in a beta distribution with an average(1)$${\;p}_{i}^{\prime }=(X_{i}+1)/(N_{i}+2)$$
and variance(2)$${SE}^{2}=p_{i}^{\prime }(1-p_{i}^{\prime })/(\;N_{i}+3).$$

Here, $N_{i}\;$ is the number of all responses in the *i-th* group, and the value $X_{i}$ is the number of those with the improved value of FA in each module. Despite employing a crude normal approximation for the distribution of differences in beta-distributed percentages, particularly when the percentage does not approximate to 50%, we can still adhere to the proof utilising the Agresti–Caffo estimate of the differences in percentages of individuals with enhanced abilities. The confidence interval for percentage differences can be expressed as follows:(3)$${∆p=(p}_{1}^{\prime }-p_{2}^{\prime })\pm z\cdot \sqrt{\frac{p_{1}^{\prime }(1-p_{1}^{\prime })}{\;N_{1}+3}+\frac{p_{2}^{\prime }(1-p_{2}^{\prime })}{N_{2}+3}}$$

This method also yields an interval estimate of the differences between the percentages of positive changes of individual clients ∆p when the sample is very small.

## 4. Statistics and Findings

### 4.1. The Values of Parameters of the Proposed Integrated Care Services

In the first step, we calculated the impact of integrated services, which included basic daily tasks, supportive daily tasks, nursing care services, and services to maintain independence. In the second step, we studied the impact of integrated services on maintaining independence separately, with the main improvement introduced according to the expected new LTC Act. From the sample of 132 older adults, we chose 75 users who received all 16 services from the group of integrated services for maintaining independence. The control group consisted of 57 LTC users without the services listed in the project MOST, as exposed in Table 1. The differences were tested in the same way as presented in formulas (1) to (3).

Following the differences in Table 1 and results in Table 5, we may conclude that, for all modules but M2, the users’ M1-M8 abilities are improved by an inventive approach to LTC at home, including reactivation services, with *p* < 0.025. The Z + 4 test is used in Table 5 to estimate the interval for the percentage of people who can improve their FA under the new treatment. In Table 5, we expose the lower limit.

### 4.2. Disparities Between the Suggested Integrated Care Services’ Parameters and Those of Previous Methods

We evaluated the hypothesis that among older people eligible for long-term care, there are disparities in the success distribution between those who received an innovative approach to integrated LTC and those who received the traditional form of treatment. Table 6 presents the findings.

The *p*-value in the chi-square test for data presented in Table 6 equals 1.64 × 10^−6^. The number of cases with increased (+) FA after innovative treatment was 37 or 49.3%. Using the Agresti–Caffo Z + 4 test, in this case, we calculate ${\;p}_{1}^{\prime }$ as follows:$${\;p}_{1}^{\prime }=\frac{X_{1}+1}{N_{1}+2}=\frac{37+1}{75+2}\;=0.49;\;SEp_{1}^{\prime }=0.0566;z=-0.49/0.0566=8.719$$

The results indicate that FA conditions could be better improved through the innovative approach of integrated care rather than solely preserved by standard home care, as evidenced by *p* < 0.0001.

Furthermore, comparing the new and old methodologies reveals substantial differences in the sample’s increase in FA. The Agresti–Caffo Z + 4 test was employed to analyse the difference in the percentage of potential successful improvements between integrated and standard home care in rural municipalities.
$${\;p}_{2}^{\prime }=\frac{X_{2}+1}{N_{2}+2}=\frac{6}{59}=0.10;\;SEp_{2}^{\prime }=0.31$$
$$SE=\sqrt{\frac{p_{1}^{\prime }(1-p_{1}^{\prime })}{\;N_{1}+3}+\frac{p_{2}^{\prime }(1-p_{2}^{\prime })}{\;N_{2}+3}}=\sqrt{\frac{0.49\cdot 0.51}{78}+\frac{0.10\cdot 0.90}{60}}=\sqrt{0.0032+0.096}=0.315$$

The difference  ∆p in the percentage of successful improvements between integrated home care and standard programmes of home care for the older adults entitled to LTC is given in the interval where z represents a value from the standard normal distribution.∆p=39% ± z·31.5%

The evaluation of differences presents challenges; specifically, $\mathrm{SE}p_{2}^{\prime }$ is substantial due to the low value of $p_{2}^{\prime }$, which necessitates taking the beta distribution into account. The Z + 4 test was crucial. We may conclude that the new reactivation and other services to strengthen and maintain independence facilitate a transition to a lower category of LTC within the system of different states of abilities, and the new approaches with integrated home care are much more successful. The probability that the new strategy will outperform the previous one for an arbitrarily selected senior eligible for LTC exceeds 0.89. Fewer than 11 out of 100 seniors eligible for LTC treated with the old standard methods will receive equivalent or enhanced benefits.

### 4.3. Testing the Differences Between Results of the Old and New Approaches Only About Services to Maintain Independence

It is essential to calculate the effectiveness of integrated services in maintaining independence separately. Among the 132 participants in the MOST programme, 75 users engaged in these services (Table 7). In a study involving 75 LTC users in the MOST programme, 25 participants improved their independence scores. In contrast, only 5 out of 57 LTC users in the control group achieved enhancements. The *p*-value of 0.025 indicates that integrated services aimed at maintaining independence in integrated care within rural areas enhance independence for more than 23% of users. In contrast, the lower limit for the percentage of users experiencing improved independence in the control group is below 3% of the population.

Through the Z + 4 test, we demonstrated that the new reactivation services enable at least 23% of users to move to a lower category of LTC within the FA multi-state system, with a probability exceeding 0.975.

We have also prepared Table 8 and Figure 3 with frequency distributions by individual modules to facilitate an overview of improvements and deteriorations in conditions and preserved states of functional abilities.

Based on the previous explanations of the weighting of functional abilities presented in Table 2, the individual’s condition shifts are evident in Table 8. The greater the deviation from the mean, the greater the difference in improvement or deterioration shown by the condition assessment, as is the difference between the first FA assessment and the last FA assessment.

As in Table 8, Figure 3 also shows shifts between deteriorations and improvements in FA by individual modules, as a result of the difference between the first assessment and the last assessment, shown in the graph.

## 5. Discussion

The World Health Organization recommended in 2020 that all countries should have integrated long-term care strategies to support their older populations better. To mitigate functional and cognitive decline, it is essential to enhance the individualised comprehensive assessment of health and the physical and mental functional status of older adults across various levels of care and identify optimal strategies for improvement. For these purposes, in addition to qualitative analyses, it is also necessary to conduct in-depth and more detailed quantitative studies, which have not been carried out in MOST or any other attempts in Slovenia or the EU. Therefore, the first author, who is the MOST project manager, and her coauthors here added quantitative results of innovative approaches to LTC to the original MOST report presented in Section 4.

The complexity of health issues and the functional and social requirements of older adults, which have significant economic implications, have highlighted the necessity for individualised and more effective strategies to promote healthy ageing, enhance fall prevention, and incorporate all bio-psychosocial aspects [75,76].

Establishing a legal framework for services is essential to accomplish this in public LTC programmes. Therefore, Slovenia opted to assess the efficiency and effectiveness of delivering appropriate and sustainable services and evaluate the outcomes of these services before implementing new legislation in the LTC sector, but without quantitative assessments of the key activities, which would also allow for an evaluation of the impact on reducing the category of care and, thus, lower health fund expenditures. Independent testing of innovative methods in rural, semi-urban, and urban environments corroborated this success across three distinct areas [55]. We assessed integrated and standard services in the rural municipality of Krško, and the first author also managed it.

Until now, individuals residing in isolated rural areas have had significantly diminished opportunities to obtain professional home care services due to the high costs of caregiver travel and considerable delays. Compared to urban residents, this may have led to premature admission to a nursing home and increased costs for social insurance. In areas with significant user dispersion, it is essential to ascertain which therapies most effectively boost the quality of life. This affects the structure of services in LTC [74] and has implications for long-term care costs and insurance expenditures.

The findings of MOST contributed to improved legislation regarding LTC. At the same time, the results in Section 4 can help to evaluate the economic advantages, especially the impact of the novelty on the LTC expenses, and to improve the bylaws. What factors might render the outcomes in rural locations unique?

The main findings of the sampling, which can contribute to the economic evaluation of new approaches in LTC in further research and, thus, to the optimization of the services presented in our study, are as follows:

For all abilities but M2, the users’ M1-M8 abilities are improved by the MOST approach to LTC at home, including reactivation services, with *p* < 0.025. Table 6 indicates that M1—Ability to move in the home environment—facilitates mobility for HC users within their housing unit, and M7—Ability to be active outside the home environment—was the most successful. More than 17% of beneficiaries will improve their abilities and reduce their care category, while M2—Cognitive and communication skills—were not much improved. With a probability exceeding 0.975, the reader can assume that Module M2, on cognitive and communication skills, is the sole module where the probability of success may be zero.Comparing the cases with decreased, neutral, and increased functional abilities after treatment between old and innovative approaches, we found that the % of cases with increased abilities after innovative treatment was nearly 50%. At the same time, the old LTC method achieved progress in only 10% of patients.We also tested the differences between the results of the old and new approaches regarding integrated services to maintain independence. Among the 132 participants in the sample of the MOST programme, 75 users were examined in these services. The interval assessment of the percentage of LTC users for whom the integrated services maintain independence was between 23,27% and 44.26% in more than 97%—compared with the control group, where the autonomy rose only to between 2.52% and 17.82%.

The project’s findings indicate that collaborative efforts from diverse professions, such as medicine, nursing, social care, dietetics, physiotherapy, occupational therapy, psychology, kinesiology, and community support, are crucial for mitigating functional decline and frailty in the elderly population. The municipality of Krško has achieved positive outcomes from implementing the pilot LTC services owing to the active involvement of stakeholders at all levels. Although the doctor was from an external organisation, the teams providing health and social services at home were able to communicate directly with the user’s physician via phone if necessary. They were able to promptly inform him of any changes in the health status of the long-term care user or address his specific needs. The partnership between the health centre’s visiting staff and the medical technicians of the pilot health services was also emphasised. The significant involvement of volunteers and unpaid caregivers in home care activities was also considered. The elements mentioned are critical to understand and closely examine when discussing the impact of the pilot services.

However, one essential issue that the MOST project and our study did not examine is the impact of the built environment, especially the family residence itself, on the advantages or disadvantages of home care. Many studies have shown that the safety of living in community LTC facilities is greater than in a care home, so this aspect will also need to be examined when implementing home-based measures [77,78].

## 6. Conclusions

Integrated long-term care is a concept that has been gaining ground in recent years, not only in Europe, but also on other continents [79,80]. This concept generally includes various types and categories of care, even palliative care [81].

The Slovenian implementation of the modified German NBA tool, named NBA-SLO tool, designed for testing in the pilot project to facilitate the transition to a unified, integrated long-term care system, has demonstrated success and, it is included in Slovenian legislation. The NBA-SLO tool prioritises the user perspective, highlighting the importance of user participation throughout the planning process, from the life story to the personal plan. The procedure’s complexity, centred on each older person in Slovenia, necessitates increased financial resources. These expenditures encompass not only the assessment implementation but also the training of experts with advanced interdisciplinary expertise.

Effective LTC care requires, of course, good organisation of programme implementation. In addition to the effective organisation of LTC providers along pathways of their daily workload, which is the subject of our further work, it is necessary to determine the optimal frequency of services according to their effectiveness in individual care categories. Our data show that with *p* < 0.0001, we can conclude that the creative integrated care approach developed by the MOST project can maintain and improve FA conditions. Significant progress in person-centred care was observed in the person’s “ability to move in the environment where he lives—M1” and “ability to be active outside the home environment—M7”. However, the precise differences in reflection based on the category of the care recipient have not yet been studied. The results thus far are promising, indicating that home care recipients involved in the integrated services of the MOST project can anticipate improved outcomes. The data indicate that long-term care residents receiving integrated care services were more likely to transition into lower categories of ADL and IADL status, irrespective of their initial functioning levels. This requires careful consideration of norms and standards when writing legislation for financing long-term care insurance.

However, technology is also developing rapidly, and legislation must follow suit. Mobile collaborative intelligent nursing robots have garnered considerable interest in the healthcare sector as a novel solution to the challenges posed by the growing aging population and constrained medical resources [82,83]. Further research should offer a thorough overview of advancements in this area, including home care and rehabilitation assistance aspects, and study the impact on insurance costs and related legislation. These results will be further tested in some countries in the ADRION region, continuing the joint project [79].

The extent to which these activities and new possibilities reduce the lifetime costs of LTC will be the subject of our further research, and we will try to contribute to LTC insurance schemes, too.

## Figures and Tables

**Figure 1 healthcare-13-01187-f001:**
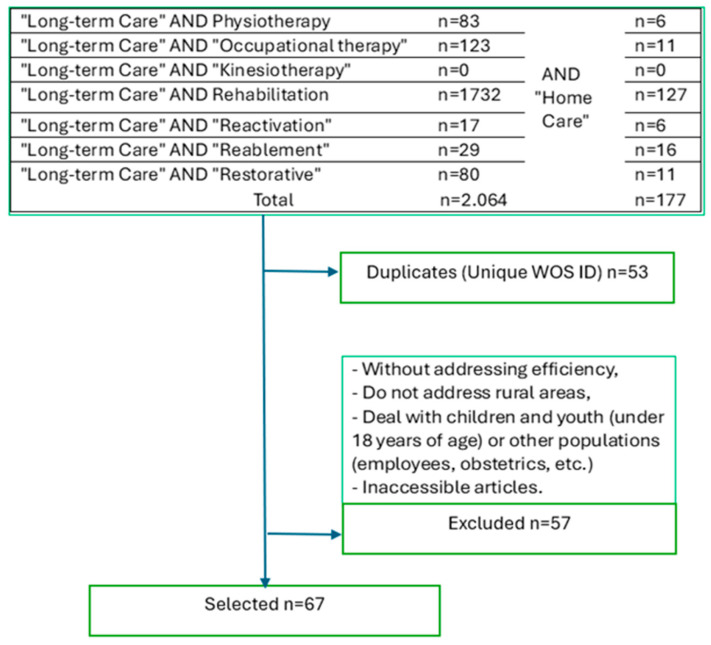
Flowchart of the Article Selection Process in Web of Science from 1990 to December 2024, based on search terms related to LTC and HC, focusing on reactivation-oriented therapies (selection conducted on 12–13 February 2025).

**Figure 2 healthcare-13-01187-f002:**
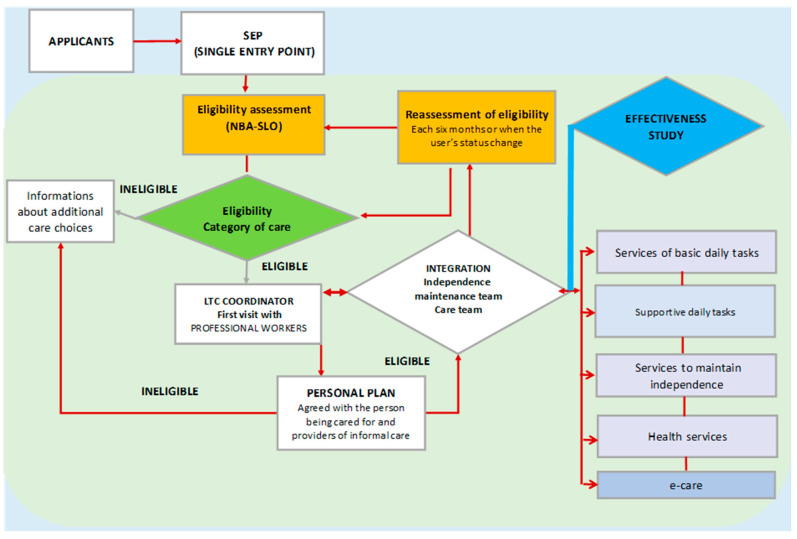
The internal process of implementing LTC in the Municipality of Krško.

**Figure 3 healthcare-13-01187-f003:**
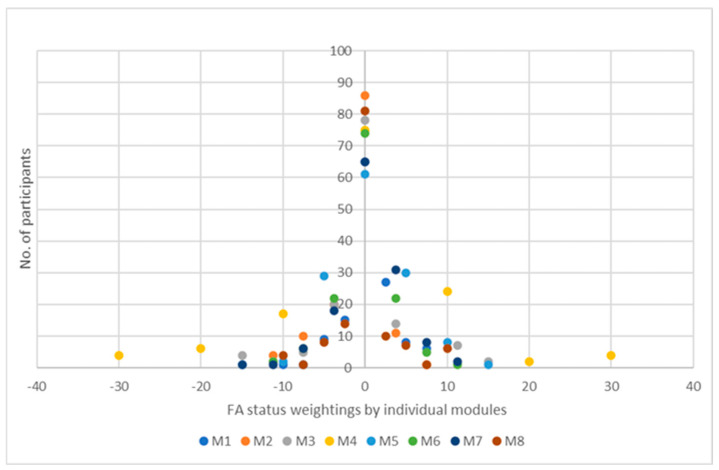
Frequency distribution of participants’ FA states, as the difference between the first and last weighted scores by individual modules.

**Table 1 healthcare-13-01187-t001:** Key differences between standard and integrated long-term care in Slovenia.

Standard Long-Term Care	Integrated Long-Term Care
Evaluation performed by LTC and HC service providers	Single entry point
Determination of the necessary category of care—only for applicants for institutional care	Determining the care category with an individual plan for all LTC applicants
Various assessment tools	The new uniform assessment tool NBA-SLO
More rights for beneficiaries in institutional care	Equalisation of the rights of beneficiaries in all forms of care
Standard non-personalised care	Person-centred care
Independent service provision by each caregiver	Teamwork
Separate social and health care services	Connected and coordinated services of different providers
Only social care in Home Care Healthcare is separated	Care team, including social, medical staff and independent maintenance team in Home Care
Services for Activities of Daily Living (ADL) and Instrumental Activities of Daily Living (IADL)	Services for ADL, IADL, physiotherapy, occupational therapy, other medical and social care, e-care

**Table 2 healthcare-13-01187-t002:** Evaluation of functional abilities by modules with five levels of functional capacity, converting into weighted point values in NBA-SLO assessment tool.

Modules	Scoring	Points/Severity Groups				
Ability to move in the environment where the insured lives	The sum of points in M1	0–1	2–3	4–5	6–9	10–15
	**Weights for M1**	0	2.5	5	7.5	**10**
Cognitive and communication skills	The sum of points in M2	0–1	2–5	6–10	11–16	17–33
	Weights for M2	0	3.75	7.5	11.25	15
Behaviour and mental health	The sum of points in M3	0	1–2	3–4	5–6	7–65
	Weights for M3	0	3.75	7.5	11.25	15
	**Weights for M2/M3**	0	3.75	7.5	11.25	**15**
Self-care ability	The sum of points in M4	0–2	3–7	8–18	19–36	37–54
	**Weights for M4**	0	10	20	30	**40**
Ability to cope with the disease and the demands and burdens associated with the treatment	The sum of points in M5	0	1	2–3	4–5	6–15
	**Weights for M5**	0	3.75	7.5	11.25	**15**
Course of everyday life and social contacts	The sum of points in M6	0	1–3	4–6	7–11	12–18
	Weights for M6	0	2.5	5	7.5	10
Ability to be active outside the home environment	The sum of Points in M7	0–6	7–10	11–14	15–17	18–21
	Weights for M7	0	2.5	5	7.5	10
	**Weighted for M6/M7**	0	2.5	5	7.5	**10**
Ability to perform household chores in the environment where the insured lives	The sum of points in M8	0–6	7–8	9–11	12–14	15–18
	**Weights for M8**	0	2.5	5	7.5	**10**

Source: Adapted from rules on the rating scale for assessing eligibility for LTC [71].

**Table 3 healthcare-13-01187-t003:** Categories of care with functional ability thresholds in weighted points.

Categories of Care	Admission	Rating with NBA-SLO (Weighted Points)
0	Ineligible	[0–12.40]
1	Eligible for LTC services under Category 1	[12.50–26.99]
2	Eligible for LTC services under Category 2	[27.00–47.49]
3	Eligible for LTC services under Category 3	[47.50–69.99]
4	Eligible for LTC services under Category 4	[70.00–89.99]
5	Eligible for LTC services under Category 5	[90.00–100.00]

Source: Adapted from Long Term Care Act ([51], Art. 12).

**Table 4 healthcare-13-01187-t004:** Overview of testing services (S1–S16).

Services	
S1. Assessment and evaluation—interview	S9. Psychosocial support for users
S2. Team/stakeholder involvement, coordinator reporting	S10. Social inclusion support
S3. Counselling on environmental adaptations	S11. Short telephone consultation (≤15 min)
S4. Training for informal caregivers	S12. Informing formal providers (GPs, nurses)
S5. Prevention, counselling, empowerment	S13. Hospital/residential care admission support
S6. Mobility support: strength, flexibility, fall prevention	S14. Safe discharge planning
S7. Chronic disease counselling	S15. Volunteer onboarding and support
S8. Health promotion for informal caregivers	S16. Extended telephone consultation (>15 min)

**Table 5 healthcare-13-01187-t005:** Interval assessment of the percentage of LTC users for whom innovative integrated care raises the FA.

Modules	M1	M2	M3	M4	M5	M6	M7	M8
X	41	12	23	30	39	28	41	24
N	132	132	132	132	132	132	132	132
*p* = X/N	0.3106	0.0909	0.1742	0.2273	0.2955	0.2121	0.3106	0.1818
*p*′ =(X + 1)/(N + 2)	0.3134	0.0970	0.1791	0.2313	0.2985	0.2164	0.3134	0.1866
SE^2^	0.0049	0.0058	0.0057	0.0054	0.0050	0.0055	0.0049	0.0056
SE	0.0699	0.0764	0.0752	0.0734	0.0706	0.0740	0.0699	0.0750
1.96*SE	0.1371	0.1498	0.1474	0.1439	0.1384	0.1450	0.1371	0.1469
Upper limit *p*′	0.4505	0.2468	0.3265	0.3752	0.4369	0.3614	0.4505	0.3335
Lower limit *p*′	0.1764	−0.0528	0.0317	0.0875	0.1601	0.0715	0.1764	0.0396

**Table 6 healthcare-13-01187-t006:** Number of cases with decreased (−), neutral (o), and increased (+) FA after treatment.

Entitled to LTC	−		o		+		Total	
	X_-_	%	Xo	%	X_+_	%	N	%
Under integrated approach	6	8	32	42.7	37	49.3	75	100
Under standard approach	15	26.3	37	64.9	5	8.8	57	100
Total	21	15.9	69	52.3	42	31.8	132	100

**Table 7 healthcare-13-01187-t007:** Interval assessment of the percentage of LTC users for whom the integrated services to maintain independence raises the independence—compared with the control group.

Groups	Under Integrated Services to Maintain Independence	Control Group
X	25	5
N	75	57
*p* = X/N	0.3333	0.0877
*p*′ = (X + 1)/(N + 2)	0.3377	0.1017
SE^2^	0.0029	0.0015
SE	0.0535	0.0390
1.96*SE	0.1050	0.0765
Upper limit *p*	0.4426	0.1782
Lower limit *p*	0.2327	0.0252

**Table 8 healthcare-13-01187-t008:** Frequency distribution of FA states, as the difference between individual modules’ first and last weighted scores.

M1	M2	M3	M4	M5	M6	M7	M8	Weighted Ratings
								−40
			4					−30
			6					−20
		4		1		1		−15
	4	2			2	1		−11.25
1			17	2			4	−10
1	10	5			6	6	1	−7.5
9				29			8	−5
	20	20			22	18		−3.75
15							14	−2.5
26	34	31	27	32	30	26	27	- Deterioration: sum in No. of participants
65	86	78	75	61	74	65	81	0—Status preserved: No. of participants
41	12	23	30	39	28	41	24	₊ Improving totals in No. of part.
27							10	2.5
	11	14			22	31		3.75
8				30			7	5
6	1				5	8	1	7.5
			24	8			6	10
		7			1	2		11.25
		2		1				15
			2					20
			4					30
								40
132	132	132	132	132	132	132	132	Sample

## Data Availability

All data and materials used in this study were collected within the LTC pilot project MOST framework funded by the EU Commission and the Ministry of Health of the Republic of Slovenia. They are available upon reasonable request. Not all the collected data were processed within the MOST project, so we supplement the results with this article.

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
