# Peer review of "Effectiveness of New Reactivation Approaches in Integrated Long-Term Care—Contribution to the Long-Term Care Act"

_healthcare, 2025, doi:10.3390/healthcare13101187_

Round 1

Reviewer 1 Report

Comments and Suggestions for Authors

The manuscript entitled “Effectiveness of New Reactivation Approaches in the Integrated Long-Term Care – Contribution to The Long-Term Care Act”, in my opinion, is hard to understand. However, English is flawless.

Please, see the comments below.

It is recommended that authors add the complete addresses of their institutions. Moreover, please add initials within your email addresses. Please check the contributions. In my opinion, not all the author’s roles are included.

The introduction section is too long. However, the authors presented relevant information here. I want to emphasize the described problem of shorter hospital stays of older adults, translated into the necessity of appropriate home care. I am especially confused with the sentence in line 200 starting from “To obtain information on the effectiveness of care”. It would be a good idea to generate two separate manuscripts – review and original. Table 1 summarizes your search, but it is not a systematic review, which should be registered, the search date should be specified, and the flowchart should be presented. Moreover, I cannot see the keyword “older adults” or “older individuals” in your search. I guess the journal’s readers know the necessity of reliable LTC programs and the difficult situation of seniors in different countries. Besides, I would use a graph instead of a table to show the dynamics of the articles, if any. Please convince me to your presentation or make any changes to make the introduction less tiring for the reader. Or maybe please move some information to the discussion, which is very poor (indeed, it is not a discussion at all); there are only three new citations. Please consider substantial changes.

What was the mean age of the Most study participants?

The more I read, it confirms my conviction that you should write two separate papers… Maybe a protocol of study and an original article? Reading the methodology, I am not sure if this study is described or maybe the MOST study? What is the purpose of Table 2? (Maybe more suitable for a review paper?).

Dear Authors, please organize your manuscript. I don't know what your contribution here is. Conclusions are disconnected from the discussion (which is lacking here). Please explain your paper to me and clearly show your contribution. Nevertheless, this study is significant and potentially interesting to the journal’s readers.

Author Response

Dear Reviewer

Thank you for bringing the shortcomings and ambiguities to our attention. We have done our best to address all of your suggestions and observations. Several corrections and additions have been made. Our responses to your comments are provided in the attached document.

Reviewer 2 Report

Comments and Suggestions for Authors

- Methods Section
Authors should divide the section into subsections such as study design, participant selection, intervention details, assessment tools, and statistical analysis. The clear structure allows readers to more easily locate specific information.
Regarding sample inclusion/exclusion criteria: the rationale for including or excluding certain participants is explained but could be more concisely presented in bullet points or a flow diagram (e.g., a CONSORT-style flow chart). 
Simplifications of Statistical Descriptions: the explanation of the Agresti-Caffo Z+4 test may be overly technical for readers unfamiliar with advanced statistics. Authors can provide a simplified explanation or give a source of reference for readers to consult. Including a short justification for the choice of this test (e.g., small sample size, binomial proportions) would be helpful.
Intervention: Authors could more clearly distinguish which of the 16 services were evaluated in the effectiveness analysis, particularly for the subgroup receiving “services to maintain independence.” A summary table listing these services and how they were delivered (frequency, intensity) would aid transparency and reproducibility.

Results Section
The results are text-heavy and statistical tables, but they are difficult to interpret quickly. Authors can include bar charts showing improvement percentages by module, a side-by-side comparison of integrated vs. standard care outcomes, and a diagram illustrating shifts in care categories (e.g., from Category 3 to Category 2).
Statistical Significance with Interpretation: The p-values and confidence intervals are reported clearly, but their implications could be explained more narratively. For instance, what does a lower limit of 23% for improvement in the integrated care group mean for policy or practice?
Meaning of “Functional Ability (FA)” Increases: Make sure that increases in FA are defined clearly early on and consistently interpreted. Readers unfamiliar with the NBA-SLO system might not grasp the significance without clearer contextualization.

Author Response

Dear Reviewer

Thank you for bringing the shortcomings and ambiguities to our attention. We have done our best to address all of your suggestions and observations. Several corrections and additions have been made. We thank you for your kind instructions, which have helped improve the quality of our research. Our responses to your comments are provided in the document I've attached.

Round 2

Reviewer 1 Report

Comments and Suggestions for Authors

Dear Authors,

Thank you for your cooperation. In its present shape, your manuscript is better organized and easier to read. 

Best regards,

The reviewer.

Author Response

Dear Reviewer

Thank you for suggestion. I have moved the figures to the Appendix A. I have renumbered them to Figure A1, Figure A2, and all others. The changes made are evident from the document I've attached.

We remain available for further clarification and look forward to your guidance.

Kind Regards, Suzanna Mežnarec N.
